# *GNAQ* and *GNA11* Genes: A Comprehensive Review on Oncogenesis, Prognosis and Therapeutic Opportunities in Uveal Melanoma

**DOI:** 10.3390/cancers14133066

**Published:** 2022-06-22

**Authors:** Paula Silva-Rodríguez, Daniel Fernández-Díaz, Manuel Bande, María Pardo, Lourdes Loidi, María José Blanco-Teijeiro

**Affiliations:** 1Fundación Pública Galega de Medicina Xenómica, Clinical University Hospital, SERGAS, 15706 Santiago de Compostela, Spain; lourdes.loidi.fernandez@sergas.es; 2Tumores Intraoculares en el Adulto, Instituto de Investigación Sanitaria de Santiago (IDIS), 15706 Santiago de Compostela, Spain; daniel.fernandez.diaz@rai.usc.es (D.F.-D.); manuelfran.bande@rai.usc.es (M.B.); mariajose.blanco@usc.es (M.J.B.-T.); 3Department of Ophthalmology, University Hospital of Santiago de Compostela, Ramon Baltar S/N, 15706 Santiago de Compostela, Spain; 4Grupo Obesidómica, Instituto de Investigación Sanitaria de Santiago (IDIS), CIBEROBN, ISCIII, 15706 Santiago de Compostela, Spain; maria.pardo.perez@sergas.es

**Keywords:** uveal melanoma, mutation, *GNAQ*, *GNA11*, oncogenesis, prognostication, therapies

## Abstract

**Simple Summary:**

The development of uveal melanoma is a multifactorial and multi-step process, in which specific and recurrent mutations arise early. Among the recurrently mutated genes, *GNAQ* and *GNA11* are involved in the process of carcinogenesis and are mutated in 80–90% of these tumours. Historically, there has been speculation as to whether these two genes are involved in the progression of primary uveal melanoma to metastatic disease, in addition to the oncogenic process itself. For this reason, both genes have been the subject of multiple research studies. Additionally, due to their high mutation rate in uveal melanoma, these genes and the downstream signaling pathways in which they are involved have been postulated as interesting therapeutic targets. This review aims to provide a current comprehensive view of what we know about *GNAQ* and *GNA11* genes on oncogenesis, prognosis and therapeutic opportunities in uveal melanoma.

**Abstract:**

The *GNAQ* and *GNA11* genes are mutated in almost 80–90% of uveal melanomas in a mutually exclusive pattern. These genes encode the alpha subunits of the heterotrimeric G proteins, Gq and G_11_; thus, mutations of these genes result in the activation of several important signaling pathways, including phospholipase C, and activation of the transcription factor YAP. It is well known that both of them act as driver genes in the oncogenic process and it has been assumed that they do not play a role in the prognosis of these tumours. However, it has been hypothesised that mutations in these genes could give rise to molecularly and clinically distinct types of uveal melanomas. It has also been questioned whether the type and location of mutation in the *GNAQ* and *GNA11* genes may affect the progression of these tumours. All of these questions, except for their implications in carcinogenesis, remain controversial. Uveal melanoma has a distinctive genetic profile, and specific recurrent mutations, which make it a potential candidate for treatment with targeted therapy. Given that the most frequent mutations are those observed in the *GNAQ* and *GNA11* genes, and that both genes are involved in oncogenesis, these molecules, as well as the downstream signalling pathways in which they are involved, have been proposed as promising potential therapeutic targets. Therefore, in this review, special attention is paid to the current data related to the possible prognostic implications of both genes from different perspectives, as well as the therapeutic options targeting them.

## 1. Introduction

Uveal melanoma (UM) is the most common intraocular malignancy in adults and the second most prevalent type of melanoma after cutaneous malignant melanoma (CMM) [1,2]. This malignancy is associated with relatively high mortality secondary to metastasis, despite the good local disease control. Metastasis occurs in approximately 50% of UM patients, mostly to the liver [1,3,4]. Pathophysiologically, UM has a common embryologic origin with CMM; however, both tumours show different epidemiologic, molecular and biological features. Deeper sequencing studies of tumour specimens from UM have revealed the existence of large genetic differences between both neoplasms. Intriguingly, UM shows a significantly lower mutational rate than CMM and most other types of solid malignancies. Actually, UM lacks mutations in *BRAF* or *NRAS* oncogenes, which are typically mutated in CMM [5,6,7,8]. Instead, this neoplasm shows its own tumour-specific chromosomal aberrations and mutated genes that exhibit recurrent variants, constituting mutational hotspots [5]. In fact, it shows two sets of mutations that occur at different frequencies that should be considered as driver mutations; one group of mutations corresponds to those that promote the oncogenic process, and the other group is correlated with the prognosis of the disease [6].

The genetic analysis of tumoral samples in UM is crucial for metastatic risk prediction, as well as for patient management and follow-up. Historically, several histopathological, clinical and radiological parameters with prognostic value have been considered to evaluate the risk of metastasis. Nevertheless, the presence of specific somatic cytogenetic and genetic biomarkers can estimate more accurately the progression to metastatic disease. Among these specific alterations, cytogenetic studies highlighted that tumours with monosomy 3 (M3) or gain of 8q are associated with poor prognosis. Monosomy 3 is present in nearly 50% of UMs acting as an independent risk factor for metastasis that strongly correlates with significantly reduced disease-free survival (DFS). This chromosomal aberration has become the most significant prognostic parameter in UM [9,10]. The frequent coexistence of 8q gains and M3 is associated with higher metastatic rates than a single aberration and shows a 5-year mortality rate of 66% in cases of concomitant M3 and 8q gain, 40% in cases of M3 and 31% in cases with 8q gain [11,12].

Another recurrent genetic alteration that is strongly associated with a bad prognosis is the inactivation *BAP1* gene (encoding BRCA1-associated protein 1 and located in chromosome 3). The biallelic inactivation of this gene occurs in approximately 50% of primary UMs, combining M3 and a deleterious somatic mutation in the second *BAP1* allele. *BAP1*-inactivated UMs are at a high risk of metastasis [13]. 

In addition to those genetic alterations with strongly prognostic values, UM frequently exhibit other chromosomal aberrations and somatic mutations. Indeed, the most common detected somatic mutations in UM are located in the guanosine nucleotide-binding protein Q gene (*GNAQ*) and its paralogue guanosine nucleotide-binding protein alpha-11 gene (*GNA11*). Those two genes are mutated in nearly 90% of these tumours in a generally mutually exclusive pattern [6,8,14,15,16]. *GNAQ* and *GNA11* genes encode protein members of the q class of G-protein alpha subunits involved in mediating signals between G-protein coupled receptors (GPCRs) and downstream effectors [17]. Both genes have mainly been considered as driver genes in carcinogenesis, as they lead to constitutive activation of GPCR signalling. Mutations in *GNAQ* and *GNA11* genes arise early in UM and are even present at the very early stages corresponding to benign melanocytic lesions [16,17,18,19]. In contrast, they are reportedly not associated with overall survival (OS) in UM patients [20]. Notwithstanding, mutations in *GNA11* were found more frequently than *GNAQ* mutations in patient cohorts with metastatic UM (MUM) [21,22]. 

Since the role of *GNAQ* and *GNA11* genes in UM was discovered, there has been speculation as to whether or not they might be involved in the genetic prognosis of the disease. Additionally, due to their high mutational rate in UM, both genes have also been the subject of therapeutic studies as they act as the switch of the MAPK/ERK signalling pathway, which is constitutively active in these tumours. Currently, important efforts are being addressed to develop effective therapies to prevent metastatic disease, which is actually the principal determinant factor of a patient’s survival. Therefore, the inactivation of *GNAQ*/*GNA11* mutants has been proposed as a potential strategy to treat UM, leading to a rapid expansion of clinical trials [23,24,25,26]. 

In this review, we present and update the recent discoveries and studies about the role of *GNAQ* and *GNA11* mutated genes in the disease oncogenesis, prognosis as well as target therapeutic options for UM. 

## 2. Mutational Hotspots and Other Less Recurrent Described Mutations in *GNAQ* and *GNA11* Genes

The *GNAQ* and *GNA11* genes are located on chromosomes 9q21.2 and 19p13.3, respectively. They are paralogous genes with a sequence homology of approximately 90% and have a coding region comprising seven exons. The proteins encoded by these genes have similar molecular weights (42,142 and 42,123 Da, respectively), and consist of 359 amino acids that comprise the αq and α11 nucleotide-binding subunits of heterotrimeric G proteins [27]. 

The frequency of *GNAQ*/*GNA11* mutations is approximately 80–90% of the UM cases [18,20,21]. In the scientific literature, the frequencies of *GNAQ* mutations have been reported to range from 24.2 to 53.3% and those of *GNA11* mutations from 24.2 to 60% [20,28,29,30,31]. In previous studies, non-Caucasian populations have shown reduced mutation frequencies in these two genes, but more recent studies have shown that mutations in this population may be closer to the frequencies previously mentioned [32,33]. This could indicate that ethnic and demographic variables could play important roles that are yet to be elucidated. 

The pioneering sequencing studies of *GNAQ* and *GNA11* in UM showed an exclusive pattern of somatic mutations in both genes [17]. Subsequent studies have described isolated cases of mutations affecting these genes simultaneously [28]. Mutational hotspots in both genes have already been described in the literature, characterised by the presence of activating missense variants that exclusively affect exons 4 and 5, and more specifically, the arginine 183 (R183) and glutamine 209 (Q209) codons. The majority of *GNA11* mutations in codon 209 leads to glutamine to leucine (p.Q209L) and proline (p.Q209P) substitutions [20,29,32,34]. These mutations occur from one-base substitutions at codon 209 (CAG), with the most common substitutions of A > T (94.5%) and A > C (2.7%) [19]. Contrastingly, in the *GNAQ* gene, a one-base change at codon 209 (CAA) can predict the substitution of glutamine by leucine (A > T, p.Q209L) and proline (A > C, p.Q209P), in most cases [17,28]. In exon 5, other mutations, including p.Q209M, p.Q209H, p.Q209I, p.F228L, and p.M203V in *GNAQ* and p.Q209Y, p.E234K, p.E221D in *GNA11*, have also been described [28,30] (Table 1). 

Overall, the frequency of mutations in the exon 4 of *GNAQ* and *GNA11* genes is lower. In *GNA11*, most mutations are caused by C > T transitions at codon 183 (CGC), and CC > TT transitions at codons 182-183, which predict the replacement of arginine to cysteine (p.R183C) or histidine (p.R183H). Likewise, the few mutations reported to affect the codon 183 of *GNAQ* (CGA) are caused exclusively by G > A transitions [19]. Other mutations affecting the exon 4 of *GNAQ* are p.P170S, p.I189T, p.Q176R, and p.P193L, which achieve an overall frequency of 8.9% in some series [28] (Table 1).

Mutations affecting the codon p.Q209 trigger a complete loss of the GTPase activity, resulting in prolonged constitutive activation of *GNAQ* and *GNA11*, thereby generating permanent downstream signalling. In contrast, mutations that affect the p.R183 residue generate a more tenuous activation due to a partial loss of the GTPase activity [19]. 

### 2.1. G Protein-Coupled Receptors

G protein-coupled receptors (GPCRs) comprise the largest family of cell surface receptors encoded by the human genome. While cancer-related mutations in GPCR signalling are less common than those in receptor tyrosine kinases, comprehensive sequencing of the human cancer genome has revealed that roughly 20% of human cancer mutations are associated with altered GPCR signalling [34]. GPCRs have seven transmembrane α-helical segments spanning the plasma membrane, with an extracellular N-terminus and intracellular C-terminus. Upon the binding of the ligand to the extracellular domain, the receptor undergoes a conformational change that is transmitted to its cytosolic region. This activates a trimeric GTP-binding protein or G protein. The G protein is made up of three subunits (alpha, beta, and gamma), and both the alpha and gamma subunits contain covalently bonded lipid tails that serve as anchors for the G protein to the plasma membrane. When no signal is present, the alpha subunit is linked to GDP, rendering the G protein inactive. When activated, the alpha subunit undergoes a conformational shift, causing GDP to detach and GTP, which is plentiful in the cytosol, to quickly bind to GDP. When GTP binds to the G protein, it undergoes a conformational shift, activating both the alpha and beta gamma complexes. The activated alpha subunit dissociates from the active beta-gamma complex, and the activated target proteins communicate with additional cascade components [36].

Following activation, proteins belonging to the family of regulators of G protein signalling accelerate the G subunit’s intrinsic GTPase activity, reverting the G protein to an inactive GDP-bound state and so promoting the formation of the inactive heterotrimeric protein complex. The p.R183 and p.Q209 mutations are located in the Gq/11 proteins’ switch I and II domains, respectively. These mutations transform G proteins to a constitutively active state by decreasing their GTPase activity [37].

### 2.2. Mitogen-Activated Protein Kinase

Primary and metastatic UMs showed that high levels of phosphorylated MEK and ERK proteins occur in the absence of *BRAF* mutations [38]. It is well established that mutant GNAQ and GNA11 proteins activate phospholipase C (PLC), which converts phosphatidylinositol diphosphate to inositol triphosphate and diacylglycerol (DAG) [39]. Then, inositol triphosphate and DAG send out signalling messengers, including calcium and protein kinase C (PKC). PKC phosphorylation initiates the MAPK pathway by sequentially phosphorylating Raf, MEK1, MEK2, and ERK [40]. These proteins converge to form multiple transcription factors (TFs) that regulate cell proliferation and apoptosis. However, Mouti et al. demonstrated that MAPK activation has only a minor impact on the carcinogenic potential of *GNAQ* mutations [41]. Subsequent studies have shown that primary UM tumours have heterogeneity in MAPK activation when the *GNAQ*/*11* mutation is present [42].

### 2.3. Protein Kinase C

*In vitro* analysis using protein kinase C (PKC) inhibitors of various specificities support the concept that the mutant GNAQ protein increases UM proliferation through PKC activation. Wu et al. discovered that enzastaurin, which is a potent PKCβ inhibitor [43], induces G1 growth arrest and apoptosis in mutant *GNAQ* cell lines at considerably higher rates than in wild-type *GNAQ* and *GNA11* cell lines [44]. This effect is mediated by the reduced activation of the MAPK pathway, with a decrease in phosphorylated ERK and cyclin D1 levels. PKC inhibitors, sotrastaurin (AEB071) and bisindolylmaleimide I (BIM), have also been shown to have substantial anticancer action against UM cells containing *GNAQ* mutations. This fact indicates that PKC signalling is crucial in mediating the oncogenic effects of mutant Gq in UM [34,45]. Interestingly, the authors also found that inhibition of PKC, or its removal by shRNA, leads to a reduction in NF-κB signalling [45]. This evidence proposes an alternative method of cell proliferation independent of the MAPK pathway involving PKC activation.

### 2.4. Phosphatidylinositol-3 Kinase/Akt

In vitro investigations of the GNAQ mutant in UM cell lines demonstrated that the suppression of PI3K-alpha and P13K-beta had little effect on proliferation, indicating that PI3K is not the primary growth promoting factor [36]. However, when MEK inhibition was assayed, a rebound rise of PI3K/Akt was found, which implies that this pathway contributes to growth maintenance in the presence of MAPK inhibition [36].

### 2.5. YAP and Its Upstream Triggers

The Hippo-YAP pathway is a regulator of cell contact inhibition, proliferation and death [37]. Yu et al. investigated UM cell lines and patient tissue samples to determine whether mutations in *GNAQ* and *GNA11* lead to YAP/TAZ activation under physiological conditions. In vitro cell lines and human samples with *GNAQ* or *GNA11* p.Q209 mutations showed higher amounts of YAP dephosphorylation and nuclear localisation in comparison to cell lines with wild-type *GNAQ* and *GNA11*, confirming these findings [46]. Moreover, the deletion of YAP using shRNA decreased tumorigenesis when the GNAQ-mutant cells were injected into nude mice [46].

A parallel investigation by Feng et al. further defined the molecular mechanism of YAP activation using a *GNAQ*-p.Q209 mutant model [47]. Other recent studies have shown the importance of this signalling pathway in the initiation and progression of UM [48]. GNAQ signalling led to YAP activation through a guanine nucleotide exchange factor, Trio, and its downstream GTPases Rho and Rac. Rho and Rac are well-known regulators of actin cytoskeleton [49]. Dynamic regulation of YAP has been shown by its connection with cytoskeleton-associated proteins, which sequesters YAP in the cytoplasm [47]. Actin polymerisation mediated by Rho/Rac may then sequester angiomotin, allowing YAP to relocate to the nucleus [37].

## 3. Prognostic Value of *GNAQ* and *GNA11* Mutated Genes in Primary Uveal Melanoma

Prognostication of UM patients can be achieved by analysing the mutational tumour status. Since the initial UM sequencing studies, there has been controversy about the possible involvement of the *GNAQ* and *GNA11* genes in the prognosis of these tumours. Firstly, it was suggested that several lines of evidence indicate that *GNA11* mutations may have a stronger effect on melanocytes than those mutations in *GNAQ* [50]. The first level of evidence was based on the statement that there were significantly more *GNA11* p.Q209 than *GNAQ* p.Q209 mutations in UM metastases. These mutations were also more common in locally advanced primary tumours, specifically in those originating from ciliochoroidal region, which is *per se* a prognostically adverse feature [19]. This observation was also initially supported by studies in mice, where the Gna11 DsK7 mutation was observed to be more tumorigenic than the Gnaq DsK1 mutation regarding melanocyte growth stimulation [51].

Several years later, Van Raamsdonk et al. also speculated that the differences observed in mice may be a functional consequence of the different mutations rather than a real difference in function between GNA11 and GNAQ proteins. In contrast, these authors observed no significant differences in survival among those patients with *GNAQ* mutations and those with *GNA11* mutations in their study, even though they observed a trend toward increased survival among patients with tumours carrying a *GNA11* mutations [19], which contradicted the evidence described so far.

One year prior to this, Bauer et al. described that mutations in *GNAQ* were not suitable to predict DFS in patients with UM [21]. This was also in agreement with Onken et al. who indicated that *GNAQ* mutations have been shown to have similar frequencies at different clinical stages of UM progression, and to be independent of chromosomal aberrations, acting as an early oncogenic event [16].

In 2013, Koopmans et al. published that *GNAQ* and *GNA11* genes are not associated with patient outcome to an equal extent. They studied samples from 92 ciliary body and choroidal melanomas and correlated the mutational status of both two genes with DFS and other parameters. The authors concluded that the univariate analysis of patients with tumours harbouring mutations in *GNAQ* or *GNA11* genes was not significantly lower than in wild-type tumours. They also examined whether mutations in these genes affected the prognosis of patients with M3 tumours by using a survival analysis for changes in chromosome 3 stratified for *GNAQ* and *GNA11* mutations. Again, no significant effect on the DFS in tumours with M3 and the presence of mutations in *GNAQ* or *GNA11* was observed [20].

Subsequently, numerous sequencing studies on UM tumour samples have been published. The results were always diverse and controversial, since although both genes showed some correlation with other prognostic features or different trends in relation to OS, no specific and reproducible pattern was ever observed among the different cohorts of the patients studied (Table 2). In 2014 Griewank et al. reported a predominance of *GNA11* mutations in MUM, and a poorer disease-specific survival of *GNA11*-mutant tumours in a cohort of 30 UM patients with metastasis [22].

Later, Decatur et al. found that *GNAQ* mutations were associated with the absence of ciliary body involvement and greater largest basal diameter; both of these are defined as bad-prognosis characteristics. On the other hand, *GNA11* mutations were not associated with any of the analysed clinicopathologic features with prognostic value [58]. In the same year, Xiani Xu et al. evaluated the proportion of *GNAQ/GNA11* mutations in Chinese patients affected by UM, finding that metastasis-free survival was not significantly associated with the *GNAQ/11* mutations in the Kaplan–Meier analysis (*p*-value = 0.94). Those results indicated that *GNAQ* and *GNA11* mutations were not significantly associated with metastasis [33].

On the other hand, in 2018, Kennedy et al. described that in a patient cohort composed of 36 patients with primary UM, where 9 patients developed metastasis, 6 of them harboured *GNA11* mutations, 2 harboured *GNAQ* mutations and only 1 of them had no mutations in neither of those 2 genes. Once more, regarding these results, the authors proposed a bias towards *GNA11* p.Q209L mutations in metastatic disease [64]. However, the metastatic status was only available for a subset of the UM cases in this study. Thus, the authors could not speculate on any potential association with metastasis. Another work about the prognostic impact of *GNAQ* and *GNA11* mutations was also published by Staby et al. in 2018, whom found no significant differences in the prevalence of *GNAQ* and *GNA11* mutations between the patients with or without metastatic disease. Despite this, these authors found that *GNAQ* mutations showed a tendency to be inversely associated with progression to metastatic disease [63].

As it can be observed, no correlation between the presence of mutations in *GNAQ* and *GNA11* and patient survival could be demonstrated in most of the studies published to date. This is because, although most of them showed a trend supporting that mutations in the *GNA11* gene are likely to lead to more aggressive disease development, this tendency has rarely been accompanied by statistically significant results. Nonetheless, very recently, Piaggio et al. demonstrated, for the first time, that UM with mutated *GNA11* has worse prognosis than those with mutations in *GNAQ* (HR = 1.97 [95%CI 1.12–3.46], *p* = 0.02). They analysed the association between *GNAQ* and *GNA11* mutations with disease-specific survival, gene expression profiles, and cytogenetic alterations in 219 primary UMs from three different cohorts (124 from the Department of Ophthalmology, Leiden University Medical Center, Leiden, the Netherlands, 72 from the Laboratory of Tumor Epigenetics, Ospedale Policlinico San Martino, Genoa, Italy and 80 from TCGA-UVM). In their study, these authors concluded that *GNA11* mutated UMs have worse prognosis, and it is associated with high risk cytogenetic, mutational and molecular tumour characteristics that might be determined, at least in part, by differential DNA-methylation. Therefore, to date, this is the only published study in which statistical results support the existence of an association between mutations in the *GNA11* gene and worse disease prognosis in patients with UM [70]. Therefore, it could be assumed that the absence of statistically significant results in the remaining published studies, in which a bias towards the *GNA11* gene in terms of prognosis had been observed, could be due to a reduced number of analysed samples, or due to the insufficient follow-up time of patients.

### Possible Relation between GNAQ and GNA11 Genes in Inflammation and HLA Expression in Uveal Melanoma

In most types of tumours, immunological characteristics, such as inflammation, infiltrate, and HLA molecules expression, correlate with the prognosis of the disease. In UM, it has been described that inflammatory phenotype, which is characterised by the presence of immune cells, such as T lymphocytes, macrophages, and an increased HLA expression, is closely related to bad prognosis [6,57,58].

In 2019, van Weeghel et al. tried to investigate whether the expression of different types of HLA molecules in UM is similarly related to M3 or might be related to the *GNAQ*/*GNA11* mutational status, to the specific type of mutation (p.Q209P/p.Q209L), and whether or not mutations in those two genes are responsible for different degrees of inflammation. As result, they found no differences in the expression of inflammatory markers, such as HLA expression, or levels of infiltrating leukocyte, according to the presence of *GNAQ/GNA11* mutations. They also found no significant differences between the GNAQ/*GNA11* or p.Q209L/p.Q209P mutations for the survival of patients [71]. These observations support the idea that *GNAQ* and *GNA11* do not play a direct role in the regulation of inflammation, and that the type and location of mutations in these genes do not appear to affect the progression of UM.

## 4. Prevalence of *GNAQ* and *GNA11* Mutations in Metastatic Uveal Melanoma

Many studies had showed that *GNAQ* and *GNA11* genes exhibit similar mutations rates in primary UM; however, less is known about the prevalence and significance of mutations in these genes in MUM. Indeed, most of the published investigations regarding somatic mutations in UM focused on the evaluation of elapsed time from the initial diagnosis and treatment of the primary UM to the development of metastasis or death.

The pioneering studies on the role of *GNAQ* and *GNA11* in the prognosis of UM had reported that the distribution of *GNA11* and *GNAQ* mutations differs between primary and MUM, with a *GNA11* to *GNAQ* ratio of 0.7 in primary UM, and 2.6 in MUM [19]. Griewank et al., who also developed sequencing studies in MUM, reported that *GNA11* mutations were considerably more frequent than *GNAQ* mutations in these specimens. Additionally, they found that patients with *GNA11*-mutant tumours had poorer disease-specific survival (60.0 vs. 121.4 months *p*-value = 0.03) and OS (50.6 vs. 121.4 months *p*-value = 0.03), than those with tumours lacking *GNA11* mutations. Thus, they proposed that the survival data, combined with the predominance of *GNA11* mutations in metastasis, raises the possibility that *GNA11*-mutant tumours may be associated with a higher risk of metastasis and poorer prognosis than those tumours bearing *GNAQ*-mutants [22].

Recently, Terai et al. investigated the possible existence of a correlation between metastasis-to-death in MUM patients with *GNAQ* and *GNA11* mutations, the frequency of mutations in MUM specimens, and the commonly mutated *GNAQ*/*GNA11* genes in survival after development of systemic metastasis. They found similar rate frequencies for both genes in patients, where mutations in *GNAQ* and *GNA11* genes were observed in 44.8% and 47.1% of patients, respectively [35]. This result was consistent with what has been published for primary UM [17,19]. Furthermore, they identified that the survival of MUM patients might be predicted according to the differences in the type of mutation (p.Q209 vs. p.Q209L) rather than the *GNAQ* and *GNA11* genes themselves [35].

Subsequently, Isaacson et al. reported that in their cohort of patients, the *GNA11* to *GNAQ* ratio was 1.2. Contrary to other studies, the *GNA11*-mutated tumours demonstrated a longer average time to first metastasis (*GNA11* vs. *GNAQ*: 77.8 vs. 43.1 months) and a better OS (79.8 vs. 33.7 months). Nonetheless, they discuss that the differences between their results, and those from other investigations, could be due to the ratio of metastatic vs. primary tumour examined, and also to the small number of samples [69].

## 5. Targeted Therapy of *GNAQ* and *GNA11* Mutations in UM

Targeted therapy is a therapeutic modality that refers to the drugs designed to interfere with a specific molecular pathway that is believed to play a critical role in tumour development or progression [72]. As it was described, UM has a distinctive genetic profile and specific recurrent mutations that make them potential candidates for targeted therapy. Given that the most frequent mutations are those observed in the *GNAQ* and *GNA11* genes [17,20,62], these molecules and their downstream signalling pathways have been postulated as interesting therapeutic targets.

The growing relevance of this therapeutic strategy is now evidenced by the numerous relevant papers that regularly report the latest results and knowledge on UM-targeted therapies [73,74,75,76] (Figure 1, Appendix A)

### 5.1. Guanine Nucleotide Dissociation Inhibitors (GDI)

The main action of GDIs is to prevent the release of GDP, i.e., they inhibit GDP/GTP exchange, which leads to the G protein remaining in an inactivated state, and therefore, blocking its signalling. Two molecules stand out within this group, FR900359 and YM-254890 [77].

The first of these, FR900359, is a cyclic depsipeptide derived from the plant Ardisia crenata, which selectively inhibits Gαq, Gα11 and Gα14, and is inactive in cells lacking these proteins [78]. It was also shown that FR900359 prevents Gq-dependent ERK1/2 activation. Lapadula et al. demonstrated that the inhibition of oncogenic Gαq/11 signalling by using this molecule promotes apoptosis, induces G1 cell cycle arrest, and prevents UM cell colony formation [23]. In a similar way, Onken et al. observed that FR900359 was able to restore melanocytic differentiation, promote apoptosis, and inhibit cell proliferation and second messenger signalling in UM cells with Gαq constitutively activation [79].

On the other hand, the YM-254890 molecule is a cyclic depsipeptide isolated from Chromobacterium sp. QS3666 [80] also with inhibitory action on Gαq/11 activation. In 2019, a study led to the discovery of an analogue, YM-19, which possessed potent inhibition of Gαq/11-mediated signaling. This analogue, despite showing minimal loss of activity in comparison to YM-254890, exhibited the advantage of being synthesised more rapidly [81]. In a recent publication, it was concluded that YM-254890 was able to suppress oncogenic Gαq signalling and cell proliferation in a wide range of UM cells, regardless of primary or metastatic origin. Interestingly, it was also observed that *GNAQ* mutant cells were less sensitive than *GNA11* mutant cells to YM-254890 [82].

Despite the fact that heterotrimeric Gαq/11 G proteins have been known for decades, their selective inhibition is still considered a challenge. The reasons include the high affinity of GDP/GTP for the G-protein, as well as the fact that its high intracellular levels hinder biochemical competition, and that this “mutation-nonspecific” type of inhibitor can block both wildtype and oncogenic protein forms [83].

### 5.2. RAS/RAF/MEK/ERK/MAPK Signalling Pathway

It is well known that Gαq/11 activation leads to the stimulation of the downstream signalling pathway MAPK, which contributes greatly to UM carcinogenesis [84].

One of the most important intermediate effectors in this pathway are the MEK enzymes. Within the subgroup of MEK inhibitors, one of the best known and most tested molecules is selumetinib (MEK1/2 inhibitor). Ambrosini et al. demonstrated that treatment with selumetinib (AZD6244) regulated the expression of the genes involved in proliferation, cell invasion and drug resistance in tumour tissues of patients with metastatic *GNAQ/11* mutant UM [85]. Thus, based on this premise, a series of studies were developed. That same year, a phase II open-label randomised trial of selumetinib as monotherapy versus temozolomide in patients with advanced melanoma (uveal and cutaneous) was published without significant clinical benefit [86].

In 2014, the results of a clinical trial (NCT01143402) in patients with advanced UM were presented. In this study, selumetinib as monotherapy was compared to chemotherapy (temozolomide or dacarbazine), showing only a modest improvement in progression-free survival (PFS) and response rate, with no improvement in OS [87]. Subsequently, the efficacy of this molecule, in combination with dacarbazine, was assessed; the study (NCT01974752) was conducted in patients with MUM, and no prior systemic therapy revealed that the combination of selumetinib plus dacarbazine had a tolerable safety profile, but did not significantly improve PFS compared to the placebo plus dacarbazine. These results raise the possibility that dacarbazine limits the efficacy of MEK inhibitors in UM, making it interesting to study selumetinib in alternative combinations other than with alkylating agents [88,89].

Following the previous approach, Decaudin et al. evaluated the potential of drug combinations to increase the efficacy of selumetinib in UM cell lines and patient-derived xenograft models (PDXs) by first assessing the combination of selumetinib and dacarbazine. They observed that this chemotherapy agent did not improve the in vitro or in vivo antitumour efficacy of selumetinib, which is consistent with the results of the Carvajal et al. clinical trial. Thus, they tested other combinations of selumetinib with docetaxel (chemotherapy agent), AZ6197 (ERK inhibitor), and vistusertib-AZD2014 (mistusertib-AZD2014), with the latter two appearing to be the most effective in UM PDXs [90]. A multi-center phase Ib study of intermittent dosing of selumetinib in patients with advanced UM, not previously treated with a MEK inhibitor (NCT02768766), is currently recruiting. This clinical trial is based on the hypothesis that greater efficacy and better tolerability can be achieved by administering selumetinib in higher doses using a pulsatile dosing schedule.

Another orally administered selective MEK1/2 inhibitor is trametinib (GSK1120212). Falchook et al. conducted a trial (NCT00687622) including 81 patients with CMM, and 16 with advanced UM and. Although efficacy was mainly observed in BRAF-mutated CMM, among the trametinib-treated UM patients, two achieved 24% tumour shrinkage (one of whom had a *GNAQ* mutation), and four had stable disease for ≥16 weeks (including two who received treatment for >40 weeks) [91].

### 5.3. PLCβ/PKC Signalling Pathway

Phospholipase C beta (PLCβ) is able to activate several protein kinase C (PKC) isoforms and RasGRPs [92]. In UM, MAPK signalling depends on the specific PKC isoforms δ and ε, which activate the RAS-exchange factor RasGRP3 [93,94]. For this reason, the study of inhibitors of these molecules, either in monotherapy or in combination with other targeted therapies, is of great interest.

In an experimental study, the molecule enzastaurin was shown to decrease the expression and/or phosphorylation of several PKC isoforms including βII, ε and θ in *GNAQ*-mutated UM cells. Down-regulation of these PKC isoforms resulted in enhanced antitumour action through the induction of apoptosis and G1 cell cycle arrest on *GNAQ* mutant UM cells, compared to those wild type [44]. In the same study, the combination of enzastaurin and MEK1/2 inhibitor (AZD6244 or U0126) showed an increased antiproliferative effect. AHT956 is another PKC inhibitor with a selective effect on cells with *GNAQ/11* mutations that induces G1 cell cycle arrest [82,95].

In a similar preclinical approach, it was shown that the PKC inhibitor, AEB071 (sotrastaurin), was able to significantly reduce the viability of UM cells harboring *GNAQ* mutations through the PKC/NF-κB and PKC/ERK1/2 pathways specifically, by the inhibition of the expression of PKC isoforms α, β, δ, ε, and θ [45]. Based on these experimental results, in which sotrastaurin showed selective sensitivity against Gα mutant UM cell lines, an open-label, multicenter, phase I dose-escalation study (NCT01430416) in MUM patients was performed. Preliminary data of this study showed manageable toxicity at multiple dose levels, as well as clinical activity [96]. However, in vitro and in vivo studies demonstrated that PKC inhibitors in monotherapy were not able to induce sustained suppression of MAPK signalling, so dual therapy was evaluated. Hence, it was observed that the combination of PKC inhibitor, AEB071, and MEK inhibitor, MEK162 (binimetinib) or PD-0325901 (mirdametinib) leads to the sustained inhibition of this signalling pathway. On the one hand, in vitro studies showed a strong synergistic effect on the induction of apoptosis and proliferation arrest. Alternatively, in vivo studies showed it causing marked tumour regression in UM patient-derived xenograft [95,97].

In the same way, it was shown that AEB071, together with CGM097 (MDM2 inhibitor) showed an additive effect enhancing the anti-proliferative and apoptotic effect, as well as the inhibition of tumour growth [98]. Subsequently, a phase Ib dose-escalation study (NCT01801358), with the combination of AEB071 (sotrastaurin) and MEK162 (binimetinib) in adult patients with confirmed MUM was proposed. However, due to discontinuation of enrolment due to side effects, the phase II part of this trial was not conducted (Array Biopharma, now a wholly owned subsidiary of Pfizer, 2020). Other targeted therapy modalities combined with sotrastaurin have been tested and are described under the following headings.

A promising novel PKC inhibitor is the LSX196 molecule, later registered as IDE196 (darovasertib). An in vitro study with *GNAQ/11* mutant UM cells observed that the combination of LSX196 with trametinib (MEK1/2 inhibitor), showed a strong synergistic effect in reducing cell viability, while the combined treatment of this first molecule with VS-4718 (FAK inhibitor) reflected a more limited synergy [82].

Several clinical trials are currently underway for evaluation. A phase I/II study, (NCT03947385) designed to characterise the safety and anti-tumour action of darovasertib in patients with solid tumours harbouring *GNAQ/11* mutations, including MUM, CMM, and colorectal cancer among others, with an approximate completion date of late 2022–2023, presented in July 2021 a preliminary robust 57% 1-year OS in monotherapy and early partial responses, in combination with binimetinib (MEK inhibitor) and crizotinib (c-MET inhibitor), in the group of patients with MUM. More specifically, a partial response in 22% of cases, and tumour shrinkage in 79% was observed with darovasertib and binimetinib combination therapy (IDEAYA Biosciences, 2021).

Another clinical trial (NCT02601378), with an estimated completion date of June 2022, aims to characterise the pharmacokinetics/pharmacodynamics, tolerability, safety and antitumour activity of LXS196 as a single agent and in combination with HDM201 (siremadlin), an MDM2 inhibitor, in patients with MUM. In 2019, Kapiteijn et al. published preliminary analyses suggesting tolerable toxicity and encouraging clinical activity of LXS196 as monotherapy in these patients [99].

### 5.4. Hippo/YAP Signalling Pathway

Feng et al. demonstrated that the *GNAQ* oncogene is able to control the Hippo pathway through a cytoplasmic protein tyrosine kinase called focal adhesion kinase (FAK). They detailed that Gαq activates FAK through a non-canonical TRIO-RhoA signalling pathway, which in turn positively regulates the yes-associated Protein (YAP) by tyrosine phosphorylation of MOB1, inhibiting the Hippo kinase cascade and promoting tumour growth in the UM [100]. Therefore, given that FAK inhibition will regulate the Hippo-YAP pathway, the same study tested FAK inhibitors (FAKi), VS-4718 (PND-1186) and PF562771, and observed in vitro that UM cells showed a dose-dependent sensitivity to these molecules, and that FAKi inhibits YAP-dependent UM tumour growth.

It is well known that FAK can be stimulated when tumour cells are exposed to other types of tyrosine kinase inhibitors, implying therapeutic resistance [101]. On the other hand, it has been observed in different cancers that FAKi shows increased activity in combination with other antineoplastic drugs [102,103]. Therefore, several studies are currently being conducted along these lines. Thus, Paradis et al. demonstrated, both in cell lines and in xenograft and liver MUM models *in vivo,* that the pharmacological combination of MEK-ERK and FAKi showed a negative synergistic action on cell growth, as well as cytotoxic effects leading to tumour regression [104]. Two examples of this therapeutic strategy are the targeted therapy trial NCT04109456 (FAKi IN10012 alone or in combination with the MEK inhibitor cobimetinib) and NCT04720417 (VS-6063 FAKi defactinib in combination with the dual RAF/MEK inhibitor VS-6766).

### 5.5. PI3K/AKT/mTOR Signalling Pathway

PI3K/AKT/mTOR is a downstream signalling pathway that can also be activated and deregulated in the UM due to oncogenic Gα action. Within this pathway, mTOR inhibitor RAD001 (everolimus) is one of the most studied drugs, both alone, and in combination. It showed reduced cell line viability and significantly delayed UM growth in preclinical studies [105], whereas when it was evaluated in patients in trial NCT01252251 (everolimus in combination with the somatostatin receptor agonist pasireotide), it only showed clinical benefit in 26% of cases and dose adjustment since side effects were common [106].

Everolimus was also studied together with the PI3K inhibitor GDC0941, which synergistically increased apoptosis in several UM cell lines compared to monotherapies, and enhanced the antitumour effect of each agent alone in UM PDXs. Evaluation of the combination of everolimus with the PKC inhibitor AEB071 (sotrastaurin) demonstrated greater activity than single molecules, inducing cell death and observing tumour regression in several UM PDXs [98].

Following the strategy of these synergistic effects, the PI3Kα inhibitor alpelisib (BYL719) was studied in combination with sotrastaurin. In the first study, it was observed that the combination of BYL719/AEB071 decreases cell viability and induces apoptosis in *GNAQ/GNA11* UM cell lines, and similarly, inhibits tumour growth in vivo in a *GNAQ* mutant xenograft model [107]. In the second study, despite the observation of a safety profile, pharmacodynamic effects and antitumour activity consistent with other targeted inhibitors, the alpelisib/AEB071 combination did not show objective efficacy [108].

Finally, molecules of this pathway have been tested in combination with MEK inhibitors. AZD8055 (mTOR inhibitor) and MK2206 (AKT inhibitor) in combination with selumetinib (MEK1/2 inhibitor) showed, in preclinical studies, a synergistic inhibition of viability in *GNAQ* mutant cell lines and xenograft models, but no impact on apoptosis [85,109].

On the other side, GSK2126458 (PI3K inhibitor) was combined with trametinib (MEK inhibitor), a trial that showed a higher rate of apoptosis with the combination therapy compared to each PI3K and MEK inhibitor alone [110]. In this field, a clinical trial in patients with advanced UM aimed to assess the improvement in PFS by combining an AKT inhibitor GSK2141795 (uprosertib) with trametinib compared to trametinib alone; unfortunately, objective responses were rare, PFS did not improve, and dose reduction due to adverse effects was common [106].

### 5.6. Other Targets and Signalling Pathways

ADP rybosylation factor 6 (ARF6) is a GTPase that triggers, in the presence of the oncogenic Gαq mutation, multiple downstream signalling pathways, such as Rho/Rac/YAP and PLCβ/PKC. The pyrazolopyrimidinone compound NAV-2729 was identified as a promising direct inhibitor of ARF6, reducing UM cell proliferation and tumorigenesis in a mouse model. Therefore, ARF6 is a potential therapeutic target for patients with oncogenic GNAQ-driven UM [111].

As previously mentioned, c-MET inhibitors are also targets of study. Within this group, cabozantinib obtained, in a first trial (NCT00940225), encouraging results from a PFS and OS point of view [112]. However, in a recent study (NCT01835145), it was observed that this molecule not only showed no improvement in PFS, but also increased toxicity compared to the classical chemotherapeutics temozolomide/dacarbazine in MUM [113]. Similarly, crizotinib was suggested to prevent the development of metastases in a MUM mouse model [114]. Despite this, Khan et al., who conducted a phase II study (NCT02223819) of adjuvant crizotinib in high-risk UM, did not find a reduction in relapse rate [115].

## 6. Conclusions

In this review, we have attempted an interrelated assessment of the possible prognostic implication of mutations in the *GNAQ/GNA11* genes in UM and their putative therapeutic opportunities. It seems clear that both genes play an important role in the oncogenic process of UM, but their role in the prognosis of these tumours remains controversial.

Most published studies agree that *GNAQ* and *GNA11* genes are only involved in the early development of UM, playing a lesser role in its progression. Nevertheless, even in the absence of statistically significant results, several investigations have suggested a more aggressive course of tumours with the mutated *GNA11* gene, due to the general observation of a trend towards longer survival among patients with tumours carrying *GNAQ* mutations. Irrespective of these results, it should be noted that most of the published investigations regarding somatic mutations in UM focused on the evaluation of time from the initial diagnosis and treatment of the primary UM to the development of metastasis or death. Moreover, there is even less knowledge about the role of these mutations in the advanced state of UM, and even less, in MUM. Regarding this, the few published studies that focuse on assessing the role of these mutations in MUM observed similar rate frequencies for both genes in patients with metastasis, and suggested that the survival of MUM patients could be predicted according to the mutation (p.Q209 vs. p.Q209L) rather than the mutated gene [35]. After all the controversy throughout history, Piaggo et al. have recently statistically demonstrated, for the first time, that UM patients with mutated *GNA11* have worse prognosis than those with mutations in *GNAQ* [70].

The *GNAQ* and *GNA11* genes are involved in various signalling pathways that are essential for the proliferation of tumoral cells in UM. This, in combination with its high mutational rate of these tumours, makes them promising targets and regulators of the therapeutic response in UM. Despite the relatively good response of the primary UM to treatment, almost 50% of patients will develop metastatic disease [116,117]. Furthermore, there do not appear to be any notable differences in survival when comparing the different current MUM treatment modalities [118]. It is for these reasons that the development of new adjuvant therapies is urgently needed. In this context, direct oncogenic inhibition of Gαq/11, and its respective signalling pathways, have yielded promising results that could be considered a therapeutic alternative with great potential in UM patients carrying these mutations.

Today, there are still important limitations that hinder and slow down the progress in the development of new adjuvant therapies. Overall, it is worth noting that many studies are in the preclinical phase, that trials include a small sample size of patients, and that the absence of randomisation is common, as well as the fact that molecular knowledge on UM is mostly based on primary tumour samples, and not so much on metastatic specimens [59,105]. More specifically, alternative activation of a signalling pathway as a compensatory mechanism for inhibition decreases the effectiveness of monotherapy drugs. Hence, all of these points explain the move towards dual therapies against Gαq/11 that prevent rebound activation of such alternative proliferative signalling pathways, with the undesirable drawback of increased toxicity [76]. Similarly, certain chromosomal alterations and gene mutations were associated with resistance to the action of these targeted therapies, such as the relationship observed between M3 and decreased sensitivity to MEK inhibition [119]. The role of these alternative genetic alterations in the effectiveness of drugs targeting downstream *GNAQ/GNA11* signalling pathways could support the idea that these genes are not really involved in UM prognosis.

Ultimately, to overcome these obstacles and obtain consistent conclusions on the benefit of different targeted therapies, it will be necessary, in the near future, to develop protocolised in vivo studies in advanced UM that consider key aspects such as the individualised molecular-genetic profile of each patient. This includes more studies to investigate the functional and prognostic relevance of oncogenic mutations in *GNAQ/GNA11* genes, and more sequencing studies involving a larger number of tumour samples at different follow-up times to generate preliminary findings that will require further clinical validation.

## 7. Materials and Methods

This article was developed in the context of an intensive literature review. The considered publications were mainly searched in PubMed using key words or sentences such as the following: UM prognosis, driver mutations in UM, oncogenic mutations in UM, G-protein coupled receptors in UM, *GNAQ* gene, *GNA11* gene, MAPK signalling pathway, targeted therapy in uveal melanoma, genetic basis of UM, sequencing studies in UM. The results of that search form the basis for publishing this article.

## Figures and Tables

**Figure 1 cancers-14-03066-f001:**
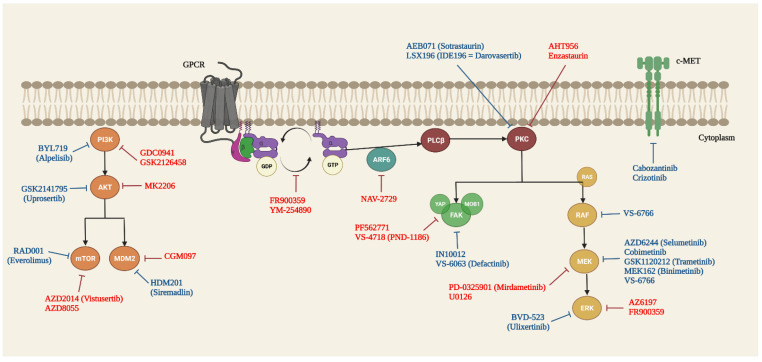
Simplified view of heterotrimeric G protein Gαq/11 and its cellular downstream signalling pathways in UM. Targeted therapies studied only preclinically are shown in red, and those already tested in clinical trials are shown in blue. Created with BioRender.com (accessed on 1 February 2022).

**Table 1 cancers-14-03066-t001:** Summary table of the different mutations described for *GNAQ* and *GNA11* genes in uveal melanoma. Chr: chromosomes.

Gene	Chr	Mutations	Exon Involved	Percentage of UM with Mutations
*GNAQ*	9q21.2	p.(Thr96Ser) p.(Pro170Ser) p.(Gln176Arg) p.(Arg183Cys) p.(Arg183His) p.(Ile189Thr) p.(Pro193Leu)	4	24.2–53.3% [17,28,30,31,32,33]
		p.(Met203Val) p.(Gln209Leu) p.(Gln209Pro) p.(Gln209Met) p.(Gln209His) p.(Gln209Ile) p.(Phe228Leu) p.(Val344Met)	5	
*GNA11*	Chr	Mutations	Exon involved	
	19p13.3	p.(Gly48Leu)	2	24.2–60% [19,28,30,31,32,33,35]
		p.(Arg166His) p.(Arg183Cys) p.(Arg183His)	4	
		p.(Gln209Leu) p.(Gln209Pro) p.(Gln209Tyr) p.(Glu221Asp) p.(Glu234Lys)	5	
		p.(Arg338His)	7	

**Table 2 cancers-14-03066-t002:** Review of the mutation ratio of *GNAQ* and *GNA11* genes in the current published UM sequencing studies and the putative prognostic implication of those genes in the disease.

	Mutation Rate		Relation with Metastasis
**Published Study**	*GNAQ* (%)	*GNA11* (%)	
Van Raamsdonk et al. (2009) [17]	46	-	-
Bauer et al. (2009) [21]			No relation between *GNAQ*-exon 5 mutations and DFS in UM (log-rank; *p*-value = 0.273).
Van Raamsdonk et al. (2010) [19]	48	34	Inverse relationship for *GNA11* p.Q209 mutations with metastatic lesions (no statistical data).
Pópulo et al. (2011) [29]	36	-	No associations between the *GNAQ* mutations and prognostic parameters.
Daniels et al. (2012) [52]	47	44	-
Furney et al. (2013) [53]	25	58	-
Harbour et al. (2013) [54]	42	52	-
Koopmans et al. (2013) [20]	50	43	No relation between patient survival in UM and mutations in *GNAQ* and *GNA11* (log-rank *p*-value = 0.466).
Martin et al. (2013) [55]	45	40	-
Dono et al. (2014) [56]	42	33	*GNAQ* is inversely associated with M3 monosomy and metastasis. Mutations in *GNA11* are related with a more aggressive tumour phenotype (no statistical data).
Ewens et al. (2014) [57]	46	35	*GNA11* mutations are positively associated with metastatic status after UM treatment (odds ratio 2.5, 95% confidence interval 1.1–5.5).
Xiaolin Xu et al. (2014) [33]	18	20	Metastasis-free survival is not significantly associated with *GNAQ/11* mutations (*p*-value = 0.94).
Johansson et al. (2015) [5]	29	50	-
Decatur et al. (2016) [58]	44	44	*GNAQ* and *GNA11* are not associated with prognosis.
Moore et al. (2016) [59]	43	49	-
Royer-Bertrand et al. (2016) [60]	58	42	-
Yavuzyigitoglu et al. (2016) [61]	49	45	-
Robertson et al. (2017) [62]	50	45	-
Kajersti et al. (2017) [63]	40	36	Mutations in *GNAQ* are inversely associated with progression to metastasis (log-rank test; *p*-value = 0.09).
Psinakis et al. (2017) [31]	18	24	No correlation between mutation status and metastasis or OS time of patients.
Staby et al. (2018) [63]	41	35	*GNA11* mutations are more frequent in the metastatic group (not statistically significative).
Kennedy et al. (2018) [64]	32	53	Suggestion of a bias towards *GNA11* p.Q209L mutations in metastasis.
Smit et al. (2018) [65]	42	44	-
Ominato et al. (2018) [32]	26	31	-
Afshar et al. (2019) [66]	58	42	No statistically significant association between M3 and mutations in *GNAQ* (*p*-value = 0.200) and *GNA11* (*p*-value = 0.200).
Piaggio et al. (2019) [67]	48	46	-
Schneider et al. (2019) [28]	20	44	Significant prolonged OS in UM with *GNAQ* exon 5 wildtype vs. mutated *GNAQ* exon 5-UM (*p*-value = 0.018) (not confirmed by multivariate analysis).
Thornton et al. (2020) [68]	53	39	
Silva et al. (2021) [30]	52	35	No correlation between mutations and metastasis or OS time (*GNAQ* log-rank *p*-value = 0.88; *GNA11* Log-rank *p*-value = 0.51).
Isaacson et al. (2022) [69]	44	52	Time to first metastasis (*GNA11* vs. *GNAQ*; 77.8 vs. 43.1 months). OS (*GNA11* vs. *GNAQ*; 79.8 vs. 33.7 months).
Piaggio et al. (2022) [70]	51.14	48.86	*GNA11* mutated UM has worse prognosis (HR = 1.97 (95%CI 1.12–3.46), *p* = 0.02).

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
