# Peer review of "GNAQ and GNA11 Genes: A Comprehensive Review on Oncogenesis, Prognosis and Therapeutic Opportunities in Uveal Melanoma"

_cancers, 2022, doi:10.3390/cancers14133066_

Round 1

Reviewer 1 Report

The authors present an excellent review of the role of GNAQ/11 mutations in uveal melanoma. This provides valuable information to both clinicians and scientists. The main detractor from the manuscript is the need for some English language grammatical editing throughout to be sure all the important points made are clear to those who may be less familiar with the subject matter. The authors could also use more active voice when writing and eliminate extra phrases that do not add meaning, such as “it has been shown that,” “it was observed that,” and “it was concluded that.”

  1. Introduction
    1. Minor grammatical error: “second most prevalent” rather than “second more prevalent”
    2. Grammar again: “despite good local disease control” rather than “even though the good local control of the disease”
    3. Patho-physiologically has no hyphen. It is just one word.
    4. “both tumours exhibited” – why change to past tense here when the introduction was in present tense?
    5. “deeper sequencing studies…have revealed” rather than “has revealed”
    6. Overall, there are grammatical issues that should be improved throughout the manuscript
  2. Section 2
    1. The list of point mutations in GNAQ and GNA11 may be better summarized in table format.
  3. Section 3
    1. The second sentence of this section should be reworded – the sentence could be more succinct and display the intended meaning more clearly.
    2. Much of the text in this section is presented in the tables. I would advise trying to minimize duplication of material in text that is in the tables. Perhaps this section could be a little shorter as a result.
  4. Please review the entire manuscript to be sure singular nouns have singular verbs and plural nouns have plural verbs. There are numerous mismatches throughout the manuscript.
  5. Section 5
    1. If this is a true review, it would be nice to have some methods stated describing the literature search strategy for studies included in this section and in the supplemental table.

Author Response

Dear referee,

Attached you can find the revised version of the manuscript entitled “GNAQ and GNA11 genes: a comprehensive review on oncogenesis, prognosis and therapeutic opportunities in uveal melanoma”. We have made changes in the main manuscript according to all the comments proposed, as well as some small corrections. As recommended, we have also improved the English.

All the modifications and improvements requested have been taken into account, including minor corrections. We also included a very recent publication (14th May 2022) about the role of GNA11 mutaions in the prognosis of UM (Piaggio et al. 2022; PMID: 35580369).

In our opinion, we have carefully addressed the reviewer’s comments. Therefore, we believe that the manuscript has been greatly improved by these modifications.

These are the answers to the referee’s comments (in bold):

  1. Introduction
    1. Minor grammatical error: “second most prevalent” rather than “second more prevalent”
    2. Grammar again: “despite good local disease control” rather than “even though the good local control of the disease”
    3. Patho-physiologically has no hyphen. It is just one word.
    4. “both tumours exhibited” – why change to past tense here when the introduction was in present tense?
    5. “deeper sequencing studies…have revealed” rather than “has revealed”
    6. Overall, there are grammatical issues that should be improved throughout the manuscript

We want to thanks the referee for the correction. We have changed the original sentences (1-5) for those recommended by the referee. Moreover, we’d like to mention that all the manuscript was reviewed in order to check the language.

  1. Section 2
    1. The list of point mutations in GNAQ and GNA11 may be better summarized in table format.

We want to thanks the referee for the advice. Therefore, we have included a new table in the text to better summarized the list of point mutations in GNAQ and GNA11. This table is called table 1 and it can be found in page 4.

  1. Section 3
    1. The second sentence of this section should be reworded – the sentence could be more succinct and display the intended meaning more clearly.

We appreciate the indication of the referee with. We have reworded the indicated sentence. Page 4, lines 236-237: "Since the initial UM sequencing studies, there has been controversy about the possible involvement of the GNAQ and GNA11 genes in the prognosis of these tumours".

    1. Much of the text in this section is presented in the tables. I would advise trying to minimize duplication of material in text that is in the tables. Perhaps this section could be a little shorter as a result.

We agree with the referee and we have reduced the material presented in the table which is better explained in the text. We just keep relevant and numerical information in the table in order to summarize the different published studies about GNAQ and GNA11 mutations. 

4. Please review the entire manuscript to be sure singular nouns have singular verbs and plural nouns have plural verbs. There are numerous mismatches throughout the manuscript.

We want to thanks the referee for the correction, all the manuscript was reviewed in order to check the language to be sure that singular nouns have singular verbs and plural nouns have plural verbs

5. Section 5

  1. If this is a true review, it would be nice to have some methods stated describing the literature search strategy for studies included in this section and in the supplemental table.

We agree with the referee. Therefore, we have included another section in the final of the article as you can see in red in page 15. 

" This article was developed in the context of an intensive literature review. The considered publications were mainly searched in PubMed by using key words or sentences such as: UM prognosis, driver mutations in UM, oncogenic mutations in UM, G-protein coupled receptors in UM, GNAQ gene, GNA11 gene, MAPK signalling pathway, targeted therapy in uveal melanoma, genetic basis of UM, sequencing studies in UM. The results of that search form the basis to publishing this article."

Reviewer 2 Report

In this Review, Silva-Rodriguez et al discussed about the role of GNAQ and GNA11 mutation in the onset and progression of uveal melanoma (UM). They specifically focused on different downstream effectors activated following by   GNAQ/GNA11 mutations and how these could be exploited as therapeutic targets for improved UV patients' prognosis.

Overall, this Review is well written and well organized.

As mentioned by the Authors, monosomy 3 is a relevant genetic event in UV, thus I would consider adding a paragraph discussing about its frequency, association with metastatic dissemination, implication in pathway activation, etc. 

Also, it would be nice to add in the conclusion/discussion the limitations of many studies due to inadequate in vitro and in vivo models and how this could be overcome in the future.

Author Response

Dear referee,

Attached you can find the revised version of the manuscript entitled “GNAQ and GNA11 genes: a comprehensive review on oncogenesis, prognosis and therapeutic opportunities in uveal melanoma”. We have made changes in the main manuscript according to all the comments proposed, as well as some small corrections. As recommended, we have also improved the English.

All the modifications and improvements requested have been taken into account, including minor corrections. We also included a very recent publication (14th May 2022) about the role of GNA11 mutaions in the prognosis of UM (Piaggio et al. 2022; PMID: 35580369). 

In our opinion, we have carefully addressed the reviewer’s comments. Therefore, we believe that the manuscript has been greatly improved by these modifications.

These are the answers to the referee’s comments (in bold):

1- As mentioned by the Authors, monosomy 3 is a relevant genetic event in UV, thus I would consider adding a paragraph discussing about its frequency, association with metastatic dissemination, implication in pathway activation, etc. 

We want to thanks the referee for the advice. Therefore, we have included in our review a new paragraph about monosomy 3 in the introduction part. This representation can be found in page 2, lines 67-86. 

2- Also, it would be nice to add in the conclusion/discussion the limitations of many studies due to inadequate in vitro and in vivo models and how this could be overcome in the future.

We totally agree with the referee. In our conclusion we said that  "to overcome these obstacles and obtain consistent conclusions on the benefit of different targeted therapies, it will be necessary, in the near future, to develop protocolized in vivo studies in advanced UM that consider key aspects such as the individualized molecular-genetic profile of each patient". 
